

# High-throughput sequence analysis reveals variation in the relative abundance of components of the bacterial and fungal microbiota in the rhizosphere of *Ginkgo biloba*

Rujue Ruan, Zhifang Jiang, Yuhuan Wu, Maojun Xu and Jun Ni

Hangzhou Normal University, Key Laboratory of Hangzhou City for Quality and Safety of Agricultural Products, College of Life and Environmental Sciences, Hangzhou, China
Hangzhou Normal University, Zhejiang Provincial Key Laboratory for Genetic Improvement and Quality Control of Medicinal Plants, Hangzhou, China

## ABSTRACT

**Background**. The narrow region of soil, in contact with and directly influenced by plant roots, is called the rhizosphere. Microbes living in the rhizosphere are considered to be important factors for the normal growth and development of plants. In this research, the structural and functional diversities of microbiota between the *Ginkgo biloba* root rhizosphere and the corresponding bulk soil were investigated.

**Methods**. Three independent replicate sites were selected, and triplicate soil samples were collected from the rhizosphere and the bulk soil at each sampling site. The communities of bacteria and fungi were investigated using high-throughput sequencing of the 16S rRNA gene and the internal transcribed spacer (ITS) of the rRNA gene, respectively.

**Results**. A number of bacterial genera showed significantly different abundance in the rhizosphere compared to the bulk soil, including *Bradyrhizobium*, *Rhizobium*, *Sphingomonas*, *Streptomyces* and *Nitrospira*. Functional enrichment analysis of bacterial microbiota revealed consistently increased abundance of ATP-binding cassette (ABC) transporters and decreased abundance of two-component systems in the rhizosphere community, compared to the bulk soil community. In contrast, the situation was more complex and inconsistent for fungi, indicating the independency of the rhizosphere fungal community on the local microenvironment.

## INTRODUCTION

Microorganisms act not only as pathogens, causing plant diseases (*Chisholm et al., 2006*), but also as neighbors or symbionts, playing important roles in plant productivity and health by providing a plethora of functional activities. Plant-associated microorganisms are often referred to as the plant's second genome and have received substantial attention in recent years (*Berg et al., 2014*; *Turner, James & Poole, 2013*). For studying the microbial diversity of prokaryotes, such as bacteria, PCR amplification of the ubiquitous 16S

Corresponding author
Jun Ni, nijun@hznu.edu.cn

ribosomal RNA (rRNA) gene, coupled with high-throughput sequencing technologies, have allowed identification of even rare non-culturable microbial species in a sample (*Caporaso et al., 2011*; *Youssef et al., 2009*). For studying eukaryotic microbes, such as fungi, the hypervariable internal transcribed spacer (ITS) rDNA sequences are often used to assess taxonomic diversity (*Bachy et al., 2013*; *Bengtsson-Palme et al., 2013*).

The narrow region of soil, in contact with and directly influenced by plant roots, is called the rhizosphere. The rhizosphere is critical to normal plant growth and development, with all the inorganic and organic substances exchanged between the root and soil occurring through this zone. On the other hand, plants also modify the rhizosphere to better adapt to the ever-changing environment (*Ryan, Delhaize & Jones, 2001*). Significant differences have been reported with respect to the physical, chemical and biological characteristics of the plant rhizosphere soil compared to the surrounding bulk soil, and this phenomenon is known as the "rhizosphere effect" (*Hartmann, Rothballer & Schmid, 2008*). Specifically, enormous numbers of microbes live in the rhizosphere, forming a complex plant-associated microbial community, and this community is considered to be crucial for plant health (*Berendsen, Pieterse & Bakker, 2012*).

*Ginkgo biloba*, a "living fossil", is an important long-lived native Chinese tree species with no living relatives (*Zhou, 2009*). It is used as a medical plant, with ginkgo herb being commonly used as an herbal dietary supplement and for the treatment of many ailments, including Alzheimer's disease (*Rimbach et al., 2001*). It is also one of the horticultural tree species which is now widely planted in China. Moreover, *G. biloba* is regarded as a valuable municipal tree in many cities in China (Fig. S1, Table S1). Due to its importance to human health and the environment, much research has been carried out on ginkgo, especially focused on the flavonoid biosynthesis pathway in ginkgo leaves (*Ni et al., 2018a*; *Ni et al., 2018b*; *Ni et al., 2017*). However, our knowledge of the relationship between ginkgo and the microbiota in its rhizosphere is still limited.

In this research, the 16S and ITS sequences were analyzed in bacteria and fungi, respectively, to estimate the diversity of microbiota in the ginkgo root rhizosphere compared to the bulk soil. A number of bacterial genera were found to accumulate in the rhizosphere, while there were also some bacterial genera for which the abundance decreased in rhizosphere. In contrast, the situation with the abundance of fungal genera in the rhizosphere versus the bulk soil was complex and inconsistent across the three sampling sites, indicating that fungal abundance was relatively independent of the rhizosphere microenvironment. Our results identified a number of bacterial genera, the abundance of which differed between the rhizosphere and the bulk soil, indicating a complex relationship between ginkgo and soil microbes.

## MATERIALS & METHODS

### Sampling and DNA extraction

Three independent replicate sites in the campus of Hangzhou Normal University were selected for soil sample collection in August 2017. Ginkgo roots were collected at a depth of about 20 cm below ground level. Large soil aggregates were removed by shaking the

roots, and the rhizosphere soil was defined as the remaining soil particles adhering to the roots. The bulk soil was collected about 10 m away from the ginkgo trees at a depth of about 20 cm below ground level. The bulk soil was also free from roots of other plants. For the collection of rhizosphere soil, roots were transferred to a 15 ml-Falcon tube containing 2.5 ml phosphate-buffered saline (PBS) solution, and were sonicated 30 times (Scientz-JY92-IIN; Scientz, Ningbo, China), each consisting of 30 s pulses at 160 W, with breaks between pulses of 30 s. After removing the roots, the suspension was centrifuged at 1,500 × g for 20 min, and the pelleted soil was used for DNA extraction. The PowerSoil$^{TM}$ DNA Isolation Kit (MoBio) was used to extract the DNA from the rhizosphere and bulk soil samples, following the manufacturer's instructions.

## PCR amplification and sequencing of amplicon libraries

The DNA samples were individually amplified by PCR using primers S-D-Bact-0341-b-S-17 (5′-CCTACGGGNGGCWGCAG-3′), S-D-Bact-0785-a-A-21 (5′-GACTACHVGGGTATCTAATCC-3′) for 16S rDNA in bacteria (*Klindworth et al., 2012*), and ITS1 (5′-TCCGTAGGTGAACCTGCGG-3′), ITS2 (5′-GCTGCGTTCTTCATCGATGC-3′) for ITS in fungi (*Sreenivasaprasad et al., 1996*). Each 30 µl PCR reaction mixture contained 5∼10 ng DNA template, 15 µl 2 × Master Mix (Phusion® High-Fidelity PCR Master Mix with GC Buffer, New England Biolabs, USA), with each primer in the reaction mixture being supplied at a concentration of 3 µM. Cycling conditions included initial denaturation at 98 °C for 1 min, followed by 30 cycles of denaturation at 98 °C, each cycle lasting 10 s, annealing at 50 °C for 30 s, and extension at 72 ° C for 30 s; a final extension phase was performed at 72 °C for 5 min. The PCR products were separated on a 2% (w/v) agarose gel. The DNA bands between 400 bp and 450 bp were collected. The DNA was extracted from the gel slices using the GeneJET gel extraction kit (Thermo Scientific, USA). The amplicon libraries were constructed using NEB Next® Ultra$^{TM}$ DNA Library Prep Kit for Illumina (New England Biolabs) according to the manufacturer's protocol. The qualified libraries were sequenced on an Illumina Hiseq2500 platform and 250 bp paired-end reads were generated. The raw sequencing data have been deposited in NCBI Sequence Read Archive (SRA) under accession number PRJNA565829 for bacteria, and PRJNA566252 for fungi.

## Sequencing data analysis, OTU production and annotation

The original paired-end reads, cutting off the barcode and primer sequences, were merged to total tags with FLASH (v.1.2.11) (*Magoc & Salzberg, 2011*). The total tags were filtered by Qiime (v.1.9.1) (*Bokulich et al., 2013*; *Caporaso et al., 2010*). These tags were then compared with the reference database (Gold database, http://drive5.com/uchime/uchime_download. html) using UCHIME algorithms (http://www.drive5.com/usearch/manual/uchime_algo. html) to detect and remove the chimera sequences (*Edgar et al., 2011*; *Haas et al., 2011*). After that, these tags were termed effective tags and were ready for further analysis.

The effective tags with ≥97% similarity were assigned to the same operational taxonomic units (OTUs) using Uparse (v8.1.1861), and the sequence with the highest frequency of occurrence in each OTU was selected as the representative sequence for further annotation

(*Edgar, 2013*). For each representative sequence, the annotation was performed using the uclust method and the Silva database to the level of kingdom, phylum, class, order, family, genus and species, to determine the community composition of each sample. OTU abundance information was normalized using a standard of sequence number corresponding to the sample with the least sequences (For bacteria, the number was 22,755. For fungi, the number was 34,819). Subsequent analyses were all performed based on these normalized data.

## Alpha and beta diversity analysis

Alpha diversity was applied to analyze complexity of species diversity for a community through four indices, including abundance-based coverage (ACE) index, Shannon diversity index, phylogenetic diversity (PD)_whole_tree and Good's coverage index. All these indices from our samples were calculated with QIIME (Version 1.9.1) and displayed using R software (Version 3.2.2). The ACE estimator (http://www.mothur.org/wiki/Ace) was selected to determine community richness, while the Shannon index (http://www.mothur.org/wiki/Shannon) was used to estimate community diversity, and Good's coverage index (http://www.mothur.org/wiki/Coverage) was used to quantify sequencing depth.

For beta diversity analysis, principal component analysis (PCA) was applied to reduce the dimension of the original variables using the FactoMineR package and ggplot2 package in R software (Version 3.2.2; *R Core Team, 2015*). Unweighted Pair-Group Method with Arithmetic Means (UPGMA) Clustering was performed as a type of hierarchical clustering method to interpret the distance matrix using average linkage, and was conducted using QIIME software (Version 1.9.1).

## Correlation analysis and KEGG functional analysis

Correlation analysis (using Spearman's rank correlation analysis) was performed based on the changes in species abundances in different communities, using CCREPE (http://huttenhower.sph.harvard.edu/ccrepe). The first 100 groups were displayed by Cytoscape (http://chianti.ucsd.edu/cytoscape-3.2.1/). To analyze the function of the microbiota, Tax4Fun (http://tax4fun.gobics.de) was used to determine the Kyoto Encyclopedia of Genes and Genomes (KEGG) estimate. The variation analysis was carried out based on the functional abundance of the samples.

# RESULTS

## Quality metrics of microbiome sequencing data from the rhizosphere and bulk soil samples

To investigate the differences in microbiota between the rhizosphere of ginkgo roots and the bulk soil, 16S rDNA amplicon sequencing for bacteria and ITS rDNA sequencing for fungi were performed on DNA samples extracted from the ginkgo rhizosphere and from bulk soil collected 10 m from the nearest ginkgo tree. Three independent replicate sites were chosen for sample collections. Within each site, triplicate samples were collected from both the rhizosphere and the bulk soil. For bacterial 16S rDNA analysis, more than 40,000

total tags were obtained in each sample except for S-2 (38,717 total tags) in site 1, and R-4 (34,905 total tags) in site 2. After trimming and filtering, more than 80% of the tags were assigned to be effective tags that were ready for further analysis (Table 1). For fungal ITS rDNA analysis, more than 40,000 total tags were obtained in each sample except for S-8 (36,779 total tags) in site 3. Similarly, more than 85% of the tags were assigned to be effective tags that were ready for further analysis. In addition, more than 98% base calls were 99% confidence (Q20) and more than 94% base calls were 99.9% confidence (Q30) for each sample with respect to both bacteria and fungi. These results indicated that the sequencing data from 16S and ITS were acceptable (Table 1).

### OTU analysis and annotation of microbiome sequences

To study the species diversities of the microbiome from the sequencing data, the annotated clean reads (taxon reads) were clustered into OTUs. An average ($\pm$ standard deviation) number of 2823.0 $\pm$ 283.7 OTUs was obtained for bacteria, with a corresponding number of 430.8 $\pm$ 146.2 OTUs for fungi (Table 2). These taxon tags were also classified to different levels of taxonomy, and three technical replicates from the same soil and site exhibited similar taxonomic patterns, which showed that the repeatability of the experiment was acceptable. Different taxonomic patterns were observed between samples from rhizosphere and bulk soil from each of the three collection sites. This indicated marked differences in the microbiota between the rhizosphere and the bulk soil (Fig. 1, Tables S2 and S3). We also investigated the ten most-abundant items with respect to different taxonomic levels of phylum, class, order, family and genus. For bacteria, the extent of variation increased from phylum to genus (Figs. S2–S6). Specifically, the frequencies of the ten most abundant genera in rhizosphere soil were quite different in the bulk soil (Fig. S6). In contrast, we found very different patterns at almost every level of taxonomy in the fungi, which indicated even greater variation in the fungal microbiota than in the bacterial microbiota when comparing communities in the rhizosphere with those in the bulk soil (Figs. S2–S6).

### Alpha and beta diversity analysis of samples

To quantify alpha diversity, the diversities of bacteria and fungi in each sample were analyzed based on the diversity indices of ACE, Shannon, PD_whole_tree and Good's_coverage. The Good's_coverage index for all the samples was greater than 0.95 for bacteria (Table S4) and greater than 0.99 for fungi (Table S5), demonstrating that the sequencing depth was acceptable. For bacteria, generally, the three indices of ACE, Shannon and PD_whole_tree were lower in the rhizosphere soil compared to that in the bulk soil. Specifically, all three indices were significantly different in site 1, with similar variation tendencies in sites 2 and 3 (Fig. 2). This demonstrated the selective effect on bacterial diversity of the rhizosphere. In contrast, the differences in these indices between rhizosphere and bulk soil were different or even opposite in different sites for the fungi. For example, all three indices decreased significantly in the rhizosphere soil from site 1, but increased significantly in the rhizosphere soil of site 3 compared to that in the corresponding bulk soil (Fig. 2). These results indicated that the effects of the ginkgo rhizosphere on the distribution of the bacteria may be different from that of the fungi.

Ruan et al. (2019), *PeerJ*, DOI 10.7717/peerj.8051

**Table 1** **The raw sequencing data and quality control of 16S and ITS.** R-1 to R-9 are samples collected from the rhizosphere. S-1 to S-9 are samples collected from the bulk soil. AvgLen, average lengths of the clean reads. Q20 and Q30 are the ratio of bases with quality value more than 20 and 30 (error rates of less than 1% and 0.1%, respectively) in the clean reads.

| 18 samples from 3 sites | | 16S | | | | | | ITS | | | | | |
|---|---|---|---|---|---|---|---|---|---|---|---|---|---|
| | | Total reads | Clean reads | Proportion (%) | AvgLen | Q20 | Q30 | Total reads | Clean reads | Proportion (%) | AvgLen | Q20 | Q30 |
| Site 1 | R-1 | 44,196 | 38,089 | 86.18 | 413 | 98.63 | 95.17 | 44,032 | 43,394 | 98.55 | 226 | 99.83 | 99.37 |
| | R-2 | 41,143 | 34.371 | 83.54 | 413 | 98.66 | 95.28 | 40,872 | 38,929 | 95.24 | 225 | 99.82 | 99.30 |
| | R-3 | 43,319 | 37,691 | 87.01 | 413 | 98.62 | 95.10 | 40,753 | 39,207 | 96.21 | 232 | 99.81 | 99.28 |
| | S-1 | 44,224 | 39,050 | 88.30 | 419 | 98.52 | 94.76 | 43,791 | 42,005 | 95.92 | 249 | 99.54 | 98.44 |
| | S-2 | 38,717 | 34,300 | 88.59 | 419 | 98.53 | 94.74 | 44,216 | 40,097 | 90.68 | 254 | 99.55 | 98.49 |
| | S-3 | 41,748 | 37,364 | 89.50 | 419 | 98.54 | 94.82 | 40,967 | 35,942 | 87.73 | 262 | 99.55 | 98.45 |
| Site 2 | R-4 | 34,905 | 30,760 | 88.12 | 417 | 98.45 | 94.54 | 44,226 | 39,959 | 90.35 | 252 | 99.75 | 99.10 |
| | R-5 | 41,833 | 36,680 | 87.68 | 416 | 98.55 | 94.93 | 42,358 | 40,274 | 95.08 | 246 | 99.72 | 99.00 |
| | R-6 | 43,612 | 38,426 | 88.11 | 414 | 98.47 | 94.70 | 43,686 | 42,656 | 97.64 | 253 | 99.68 | 98.86 |
| | S-4 | 44,197 | 39,884 | 90.24 | 417 | 98.50 | 94.65 | 42,357 | 38,430 | 90.72 | 266 | 99.54 | 98.43 |
| | S-5 | 42,674 | 37,796 | 88.57 | 418 | 98.51 | 94.67 | 41,487 | 36,392 | 87.72 | 265 | 99.54 | 98.44 |
| | S-6 | 41,380 | 36,649 | 88.57 | 418 | 98.53 | 94.79 | 44,379 | 41,628 | 93.80 | 252 | 99.67 | 98.84 |
| Site 3 | R-7 | 40,526 | 33,739 | 83.25 | 415 | 98.58 | 95.04 | 43,107 | 42,305 | 98.14 | 234 | 99.78 | 99.19 |
| | R-8 | 40,593 | 34,748 | 85.60 | 415 | 98.55 | 94.89 | 41,706 | 39,722 | 95.24 | 247 | 99.76 | 99.13 |
| | R-9 | 41,556 | 34,434 | 82.86 | 415 | 98.46 | 94.62 | 40,768 | 37,212 | 91.28 | 238 | 99.83 | 99.35 |
| | S-7 | 42,328 | 37,973 | 89.71 | 419 | 98.54 | 94.80 | 41,338 | 41,156 | 99.56 | 258 | 99.65 | 98.80 |
| | S-8 | 42,939 | 38,061 | 88.64 | 418 | 98.47 | 94.58 | 36,779 | 35,699 | 97.06 | 261 | 99.48 | 98.30 |
| | S-9 | 43,066 | 38,837 | 90.18 | 417 | 98.48 | 94.66 | 44,097 | 42,364 | 96.07 | 261 | 99.67 | 98.86 |

Ruan et al. (2019), *PeerJ*, DOI 10.7717/peerj.8051

**Table 2  OTU clustering of sequence data.** R-1 to R-9 are samples collected from the rhizosphere. S-1 to S-9 are samples collected from the bulk soil. Taxon reads: anno-tated clean reads that are used to set up OTUs. Unclassified reads: reads without annotations. Singletons: single reads that could not be clustered to any OTUs (these reads were not subjected to further analysis).

| 18 samples from 3 sites | | 16S | | | | | ITS | | | | |
|---|---|---|---|---|---|---|---|---|---|---|---|
| | | Clean reads | Taxon reads | Unclassified | Singletons | OTUs | Clean reads | Taxon reads | Unclassified | Singletons | OTUs |
| Site 1 | R-1 | 38,089 | 30,857 | 204 | 7,028 | 2,332 | 43,394 | 42,353 | 0 | 1,041 | 452 |
| | R-2 | 34,371 | 25,898 | 150 | 8,323 | 2,216 | 38,815 | 38,013 | 0 | 802 | 340 |
| | R-3 | 37,691 | 30,274 | 216 | 7,201 | 2,433 | 39,207 | 38,570 | 0 | 637 | 327 |
| | S-1 | 39,050 | 25,265 | 789 | 12,996 | 3,001 | 42,005 | 40,290 | 0 | 1,715 | 734 |
| | S-2 | 34,300 | 22,183 | 768 | 11,349 | 2,684 | 40,097 | 38,667 | 0 | 1,430 | 531 |
| | S-3 | 37,364 | 29,709 | 930 | 6,725 | 3,241 | 35,942 | 34,819 | 0 | 1,123 | 564 |
| Site 2 | R-4 | 30,760 | 22,398 | 357 | 8,005 | 2,799 | 39,959 | 38,325 | 0 | 1,634 | 474 |
| | R-5 | 36,680 | 25,620 | 434 | 10,626 | 2,999 | 40,274 | 38,028 | 0 | 2,246 | 552 |
| | R-6 | 38,426 | 28,191 | 429 | 9,806 | 3,000 | 42,656 | 41,632 | 0 | 1,024 | 532 |
| | S-4 | 39,884 | 30,053 | 1,206 | 8,625 | 3,190 | 38,430 | 36,290 | 0 | 2,140 | 340 |
| | S-5 | 37,796 | 26,394 | 993 | 10,409 | 3,018 | 36,392 | 34,976 | 0 | 1,416 | 426 |
| | S-6 | 36,649 | 25,629 | 1,344 | 9,676 | 2,966 | 41,628 | 40,503 | 0 | 1,125 | 462 |
| Site 3 | R-7 | 33,739 | 24,797 | 243 | 8,699 | 2,695 | 42,305 | 41,409 | 0 | 896 | 556 |
| | R-8 | 34,748 | 25,363 | 323 | 9,062 | 2,795 | 39,722 | 38,696 | 0 | 1,026 | 466 |
| | R-9 | 34,434 | 24,903 | 199 | 9,332 | 2,612 | 37,212 | 36,505 | 0 | 707 | 412 |
| | S-7 | 37,973 | 30,256 | 1,777 | 5,940 | 2,837 | 41,156 | 40,657 | 0 | 499 | 244 |
| | S-8 | 38,061 | 27,511 | 1,366 | 9,184 | 3,014 | 35,699 | 35,165 | 0 | 534 | 144 |
| | S-9 | 38,837 | 27,550 | 1,795 | 9,492 | 2,987 | 42,364 | 41,909 | 0 | 455 | 199 |
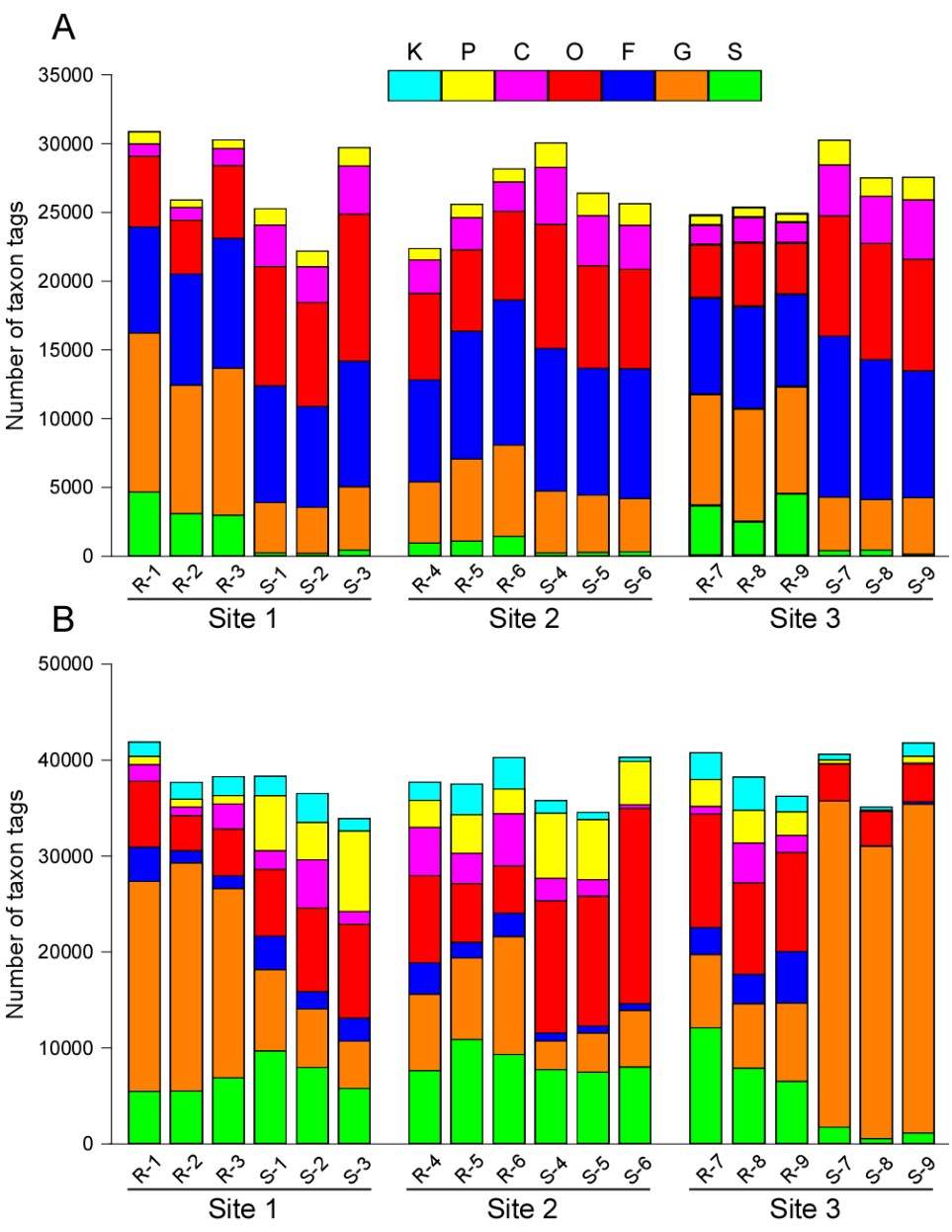

**Figure 1 Different levels of taxonomic distribution of taxon tags.** Site 1, Site 2 and Site 3 are the three sample collection sites. R-1 to R-9 are samples collected from rhizosphere. S-1 to S-9 are samples collected from the bulk soil. Different colors represent different levels of taxonomy. (A) Bacteria. (B) Fungi. K, kingdom. P, phylum. C, class. O, order. F, family. G, genus. S, species.

For beta diversity, PCA analysis was carried out to cluster bacterial and fungal communities in rhizosphere and bulk soil, according to the different sampling sites. At the OTU level, PC1 explained 19.33% and PC2 11.59% of the total variation in bacteria, and the rhizosphere samples were clearly distinguishable from the bulk soil samples (Fig. 3A). In contrast, PC1 explained 13.71% and PC2 10.05% of the total variation in

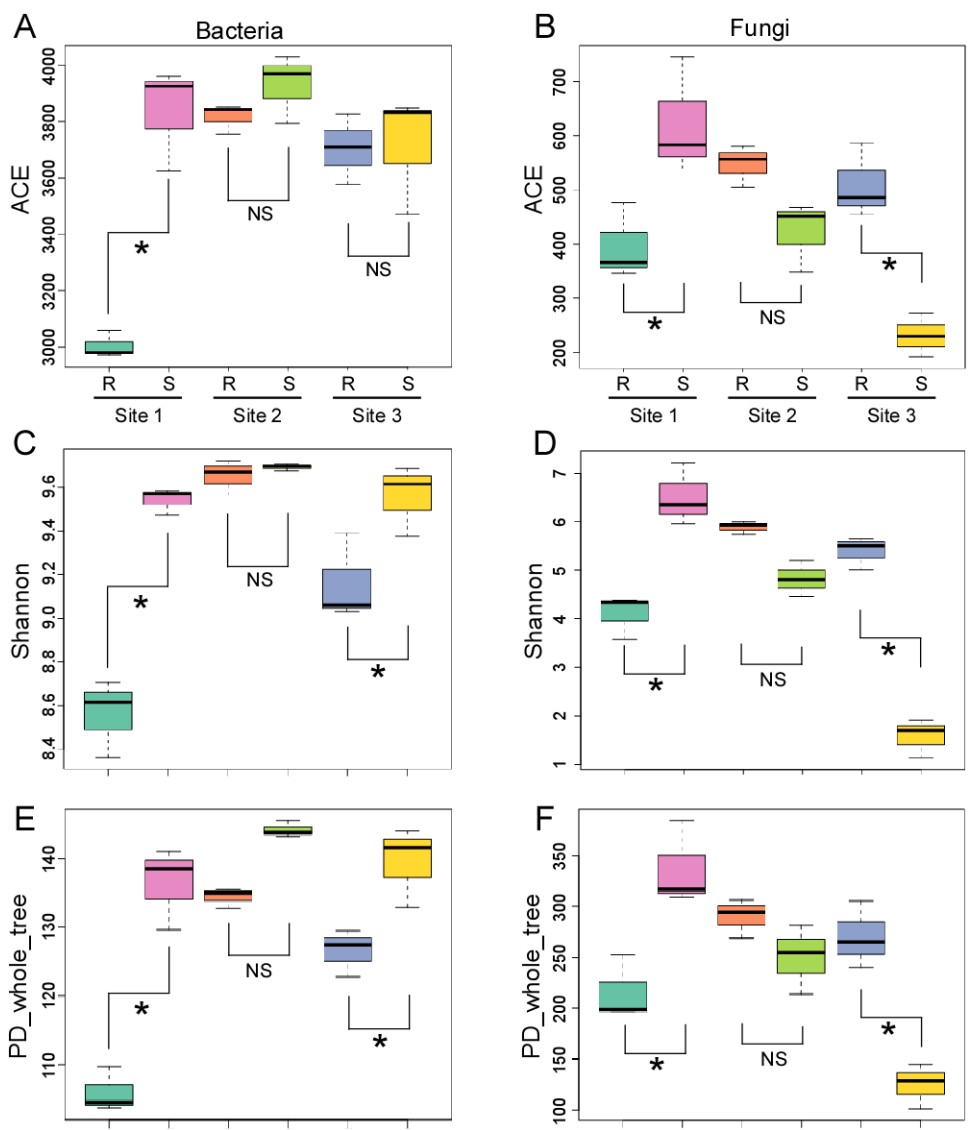

**Figure 2 Alpha diversity of the bacterial and fungal communities.** (A), (C) and (E), three indices for bacteria. (B), (D) and (F), three indices for fungi. Alpha diversity estimates represent three biological replicates for the rhizosphere (R) and the bulk soil (S) from the three sites. Significant changes ($P < 0.05$), determined by the Tukey test, are marked by an asterisk. NS, not significant.

fungi; samples from the rhizosphere and bulk soil did not separate clearly with respect to the fungi (Fig. S7A). We also performed UPGMA cluster analysis and built cluster trees for the samples. Using this approach, all the samples collected from the rhizosphere were distinguishable from the samples from the bulk soil with respect to bacteria, while the situation was less clear-cut with respect to the fungi (Fig. 3B, Fig. S7B).
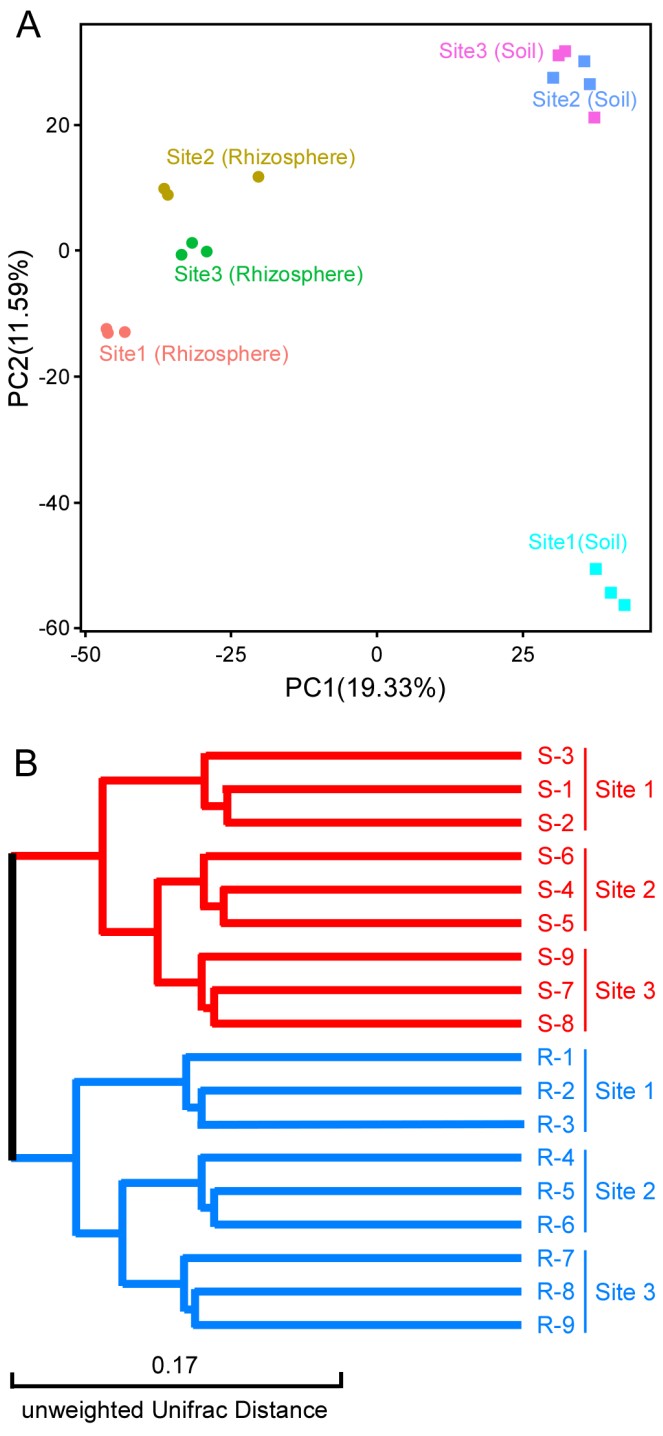

**Figure 3 Beta diversity of the bacterial communities.** (A) PCA analysis. Samples from rhizosphere and control soil are marked. (B) UPGMA cluster analysis. Samples collected from rhizosphere are marked in blue, and samples collected from the bulk soil are marked in red.

## Variations in the microbiota in the ginkgo rhizosphere at the genus level

To take a closer look at the variation in the microbiota of the ginkgo rhizosphere, the 35 most abundant genera of bacteria were compared between the rhizosphere and the bulk soil. Interestingly, the abundance of most of the genera varied between the rhizosphere and the bulk soil (Fig. 4). Several genera showed increased abundance in the rhizosphere, including *Bradyrhizobium*, *Rhizobium*, *Sphingomonas* and *Streptomyces*. In contrast, there were also some genera which showed decreased abundance in the rhizosphere compared to that in the bulk soil, including *Nitrospira* (Fig. 4). We also examined the relative abundance values in the rhizosphere and bulk soil of the 35 highest-frequency fungal genera, and found that, although some genera showed significantly different abundances between rhizosphere and bulk soil at one collection site, this was not consistent across the other sites (Fig. S8). This indicated that, for individual fungal genera, the selective effect (positive or negative) of the rhizosphere was not significant as it was in bacteria.

## Correlation and KEGG functional analysis in bacteria

Based on the changes in abundance of different genera in different soil samples, the relationships between the abundance values of the various bacterial genera were determined. It was obvious that *Rhizobium* and *Pantoea* dominated the microbiota. *Rhizobium* was the most abundant genus, and had a significant positive relationship with *Bosea*. *Pantoea* was also highly abundant and had very close relationships with many other genera, including *Bradyrhizobium* (Fig. S9). The relationship network of these genera indicated a complex functional collaboration within the microbiota. To analyze the functional diversities of bacteria, the KEGG functional enrichment analysis of bacterial microbiota was compared between the rhizosphere and the bulk soil. The frequency of ATP-binding cassette (ABC) transporters was enriched significantly in the rhizosphere while the frequency of the two-component system decreased significantly in the rhizosphere (Fig. S10). This indicated a functional divergence of bacterial microbiota in response to the rhizosphere of ginkgo roots.

## DISCUSSION

With the development of the ginkgo-based pharmaceutical industry and of ginkgo horticulture, it is increasingly important to fully understand the different aspects of ginkgo biology, including the relationship with its root microbiota. Most of the previous research on ginkgo had focused on the biosynthesis pathways of various bioactive compounds present in ginkgo leaves, which are raw materials for the pharmaceutical industry. On the other hand, little research has been carried out on ginkgo roots. To our knowledge, this research represents the first report on the relationship between ginkgo roots and the soil microbiota.

In this study, the 16S rDNA in bacteria and the ITS rDNA sequences in fungi were amplified and sequenced. We did not find the contaminant sequences from chloroplast, mitochondrial or nuclear DNA, which have frequently occurred in related research (*Beckers et al., 2016*). This finding showed that our PCR approach was optimized and suitable for

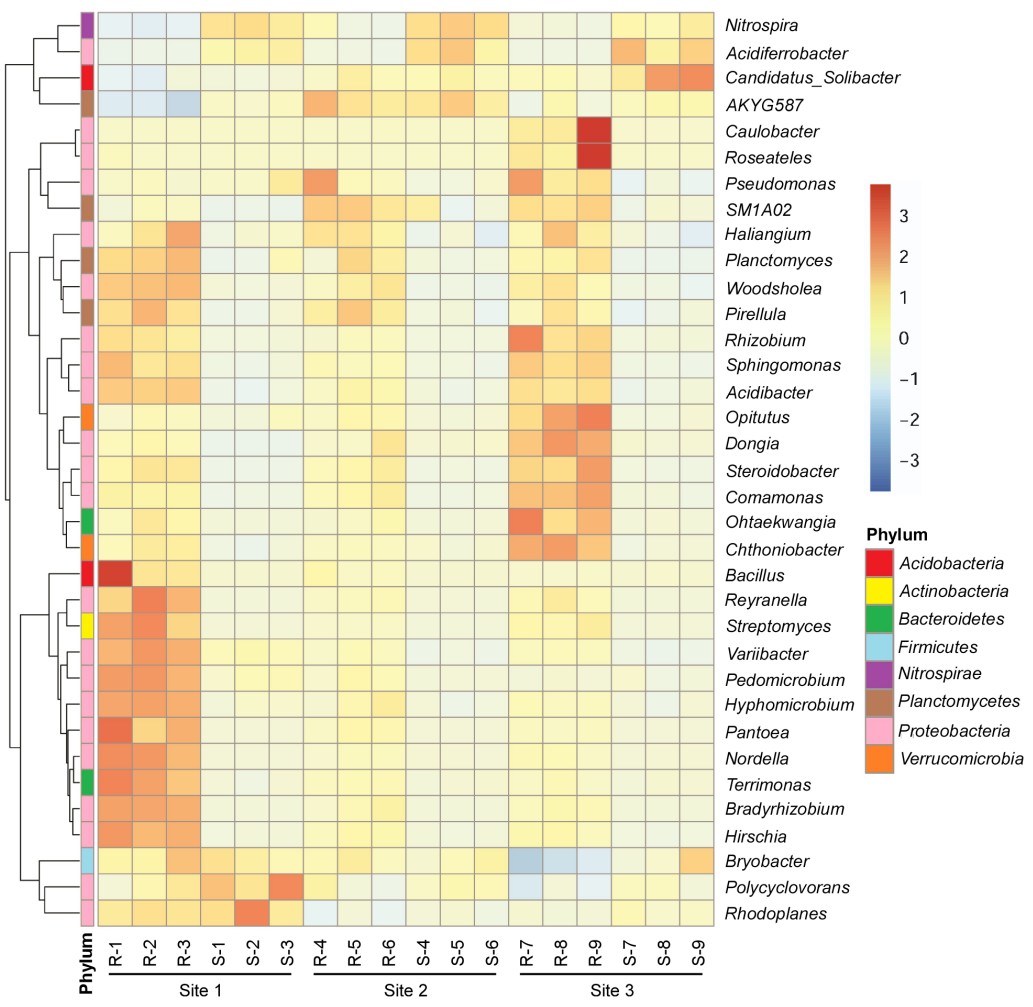

**Figure 4** **The abundance of the 35 most-abundant genera of bacteria in the rhizosphere and the bulk soil from the three sites.** The taxonomies of genera to the level of phylum are marked by different colors. The relative abundances of the 35 genera are scaled by a Z-score color gradient bar. The red colored data represent genera that have higher abundance than average. The blue colored data represent genera that have lower abundance than average. A Z-score of 0 represents a genus abundance value that is equal to the average abundance value.

this research. Considering the complex nature of the soil environment, which may cause changes in microbiota composition, we chose three different and independent sites for sample collection. Only those changes which occurred in all three sites were discussed.

One of the important findings in this study was the accumulation in the rhizosphere of species of *Rhizobium* and its fellow rhizobial genus *Bradyrhizobium*. *Rhizobium* and *Bradyrhizobium* are nitrogen-fixing bacteria which induce the development of nodules in the roots of legume plant hosts (*Long, 1996*). In addition, flavonoids are widely accepted to be regulators of symbiotic interactions, acting as specific signals between plant hosts and *Rhizobium* (*Mierziak, Kostyn & Kulma, 2014*). Considering that flavonoids are one of the most important groups of secondary plant metabolites in ginkgo (*Kleijnen & Knipschild,*
*1992*), it is possible that ginkgo root cells secrete flavonoids, which act as specific signals to attract the accumulation of *Rhizobium* and *Bradyrhizobium* in the rhizosphere. However, despite the accumulation of *Rhizobium* and *Bradyrhizobium* in the rhizosphere, we did not find any nodules on ginkgo roots. This might be due to the lack of Nod factor receptors or to defects of the subsequent kinase cascade in ginkgo, which is crucial to nodule formation in legumes (*Gage, 2004*; *Smit et al., 2007*). Following the recent publication of the draft genome of *G. biloba* (*Guan et al., 2016*), it would be interesting to identify the missing steps in ginkgo which are associated with its inability to form nodules.

We also observed an accumulation of *Sphingomonas* in the ginkgo rhizosphere. Bacteria within the genus *Sphingomonas* share the common capacity to degrade a broad range of aromatic compounds (*Fredrickson et al., 1995*). Thus, the accumulation of *Sphingomonas* in the rhizosphere indirectly suggested the secretion of different aromatic secondary metabolites from ginkgo roots, which attracted the accumulation of aromatics-consuming *Sphingomonas* species.

*Streptomyces* is the largest genus of the Actinobacteria, with more than 500 species having been described (*Euzeby, 2008*). *Streptomyces* not only produces a volatile metabolite, geosmin, which result in the distinct "earthy" odor of soil, but also produces antibiotics, which they use to compete with other bacteria for resources. A number of them have been developed as antifungals, antibiotics and chemotherapeutic drugs to improve human health (*Raja & Prabakarana, 2011*). The benefits to ginkgo of *Streptomyces* accumulation in the rhizosphere are currently unknown, but this phenomenon could increase the complexity of the composition of the microbiota in the rhizosphere.

In contrast to those bacterial genera which accumulated in the ginkgo rhizosphere, there were also some genera which decreased in the rhizosphere compared to the bulk soil. The genus *Nitrospira* consists of a group of species which are widely distributed in many natural environments (*Bartosch et al., 2002*), and they are considered to play important roles in the nitrogen cycle in both water and soil (*Hayatsu, Tago & Saito, 2008*). Despite the potential advantage of an exogenous nitrate supply to ginkgo, *Nitrospira* did not accumulate in the rhizosphere but rather was present at lower frequencies in the rhizosphere than in the bulk soil. This decline may due to the inhibitory effects of antibiotics produced by other rhizobacteria (*Streptomyces,* for instance), which accumulated in the rhizosphere. Alternatively, the decline may be caused by the complex secondary metabolites secreted by ginkgo roots. It has been reported that flavonoids, secreted by root cells, had both positive and negative effects on nodule formation by nitrogen-fixing bacteria (*Cooper, 2004*; *Khandual, 2007*). Thus, it is possible that certain compounds secreted by ginkgo root cells prevented the accumulation of certain bacteria, including *Nitrospira*.

ABC transporters are integral membrane proteins that couple the transport of substrates across lipid bilayers to the hydrolysis of ATP (*Hollenstein, Dawson & Locher, 2007*). In bacteria, ABC transporters are important factors catalyzing the uptake of nutrients and the efflux of toxic or antimicrobial agents, which are crucial for bacterial survival (*Davidson & Chen, 2004*). The accumulation in the rhizosphere of bacteria enriched with respect to ABC transporters is consistent with the function of the rhizosphere, which mediates the exchange of inorganic and organic substances between the roots and the soil

(*Ryan, Delhaize & Jones, 2001*). Two-component systems are characterized by a sensor kinase consisting of a signal-recognition domain with unique specificity, coupled to an auto-kinase domain, and they are the major means by which bacteria recognize and respond to a range of environmental stimuli (*Hoch, 2000*). For the bacteria in the rhizosphere, the frequency of two-component systems decreased compared to that in the bulk soil. We propose that bacteria must cope with different environmental stimuli in the soil. For the bacteria in the rhizosphere, the microenvironment is greatly affected by the plants. These bacteria are partially "protected" by the rhizosphere, and there is no need for these bacteria to employ numerous two-component systems to cope with the different challenges from the ever- changing environment as would be the case for the bacteria in the bulk soil.

Compared with the clear changes in bacterial frequency between the rhizosphere and bulk soil, the responses of fungal frequency between the two soil types varied between the different collection sites. Many fungal genera accumulated in the rhizosphere of one collection site, yet decreased at other collection sites (Fig. S8). Considering the different and complex subcellular structures of fungi, it is possible that the substances secreted by root cells and bacteria have relatively little effect on the distribution of the fungal microbiota.

## CONCLUSIONS

In this research, the structural and functional diversities of microbiota between the *Ginkgo biloba* root rhizosphere and the corresponding bulk soil were investigated. A number of bacterial genera showed significantly different abundance in the rhizosphere compared to the bulk soil, including *Bradyrhizobium*, *Rhizobium*, *Sphingomonas*, *Streptomyces* and *Nitrospira*. Functional enrichment analysis of bacterial microbiota revealed consistently increased abundance of ATP-binding cassette (ABC) transporters and decreased abundance of two-component systems in the rhizosphere community, compared to the bulk soil community. In contrast, the situation was more complex and inconsistent for fungi, indicating the independency of the rhizosphere fungal community on the local microenvironment. This study was the first attempt to characterize the microbiota in the ginkgo rhizosphere, which indicated a complex relationship between ginkgo and the microbial communities in the soil.

## ACKNOWLEDGEMENTS

We thank Prof. Huizhong Wang from Hangzhou Normal University for suggestions and constructive criticism of this manuscript.

### Funding

This work was supported by the Zhejiang Provincial Natural Science Foundation of China (No. LY18C060008). The funders had no role in study design, data collection and analysis, decision to publish, or preparation of the manuscript.

## Grant Disclosures

The following grant information was disclosed by the authors:
Zhejiang Provincial Natural Science Foundation of China: LY18C060008.

## Competing Interests

The authors declare there are no competing interests.

## Author Contributions

- Rujue Ruan performed the experiments, prepared figures and/or tables, approved the final draft.
- Zhifang Jiang performed the experiments, authored or reviewed drafts of the paper, approved the final draft.
- Yuhuan Wu analyzed the data, authored or reviewed drafts of the paper, approved the final draft.
- Maojun Xu conceived and designed the experiments, authored or reviewed drafts of the paper, approved the final draft.
- Jun Ni conceived and designed the experiments, contributed reagents/materials/analysis tools, prepared figures and/or tables, authored or reviewed drafts of the paper, approved the final draft.

## Data Availability

    Data is available at NCBI SRA. Bacteria: PRJNA565829, Fungi: PRJNA566252.

## Supplemental Information

Supplemental information for this article can be found online at http://dx.doi.org/10.7717/peerj.8051#supplemental-information.

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
