# Peer review of "High-throughput sequence analysis reveals variation in the relative abundance of components of the bacterial and fungal microbiota in the rhizosphere of Ginkgo biloba"

_PeerJ, doi:10.7717/peerj.8051_

## Round 0.1 · original submission · Major Revisions

Bot reviewers found merit in your work. However, they provided several comments and suggestions to improve your manuscript. I encourage authors to review the manuscript accordingly

Reviewer 1 ·

Basic reporting

The manuscript describes microbiota of Ginko bilboa rhizosphere and comparison of that with soil microbiota from a similar site. The used language is relatively good. However, field background, context and references used are very limited. In fact, almost no relevant information on previously published rhizosphere microbiota research is provided.
Raw data is shared, the structure of the manuscript is relevant and figures relatively well designed.
With a deep appreciation of the importance of Gink bilboa as widely accepted components of ethnopharmaceutical remedies, the reason why only this one species of plant from only one particular site has been chosen as study object reminds hard to understand.

Experimental design

Experimental design of laboratory work is standard and acceptable, no novel approaches are described.
Sample collection design is very limited and at this moment on an unacceptable low level.

Validity of the findings

Provided findings are relevant per se. However, the novelty of this information is hard to find.
The experiments should have been performed at least in several sites with different soil to give a deeper understanding which specific finding are ginko specific and which soil/site specific. An alternative possibility to make provided findings as meaningful would have been a comparison of several plant species in similar soil type to reveal which findings are in fact gink specific.

Additional comments

This is a technically nicely performed work, that however, does not provide any relevant new knowledge on rhizospehere microbiota as such or specifically ginko rhizosphere microbiota.

·

Basic reporting

no comment

Experimental design

no comment

Validity of the findings

See general comments for minor stats problem.

Additional comments

This paper characterizes the microbial communities associated with the rhizosphere and bulk soil in proximity to Ginko trees. In general, the science is done correctly – the methods are appropriate and the data are fairly convincing in this relatively straightforward study. However, there are multiple presentational problems that need to be addressed according to my comments below, mostly with the figures and tables. Additionally, there are some minor problems with English usage/grammar throughout that need to be addressed – I commend the authors that the manuscript is fairly good, just needs one more round of careful proofreading (esp. for certain turns of phrases).

Specific comments

Throughout: Your scientific names for Ginko and bacteria are missing their proper italics in certain places.

Abstract: In contact with, not “in touch with”. Abundances are simply different and not “altered” unless you have experimentally altered something.

1) L.93-96: If you did not design these primers yourselves, you should give their names and the references from which they come. If you did design them, you should still give names and coordinates/locations for them on the molecules so that people can compare.
2) L.122-124: It is important you state what this number is you are normalizing to.

Table 1: A rate is a change in a value over time, so you should be using proportion or simply percentage. Q20 and 30 values are not error rates “less than”, they are exactly 99 and 99.9% confidence in the base calls.

Table 2: Some of your lines don’t add up, such as R-1 that has 30854+204+7028=38086 and not 38089. You also have inconsistencies in the table numbers with commas missing and some periods instead.

General note: Some of the below comments are made about the main figures, but are also applicable to some of the Supplementary Figures and you should adjust them accordingly.

Figure 2: Unless your values are all normally distributed (you should run test), then you should not be using the t-test, but the non-parametric equivalent (Mann-Whitney).

Figure 3: I would not use the arrows in panel A (use a legend or just labels next to them with circles perhaps) as these types of plots can often have vectors on them and these arrows lend one to think of that mistakenly. In panel B, red and green are not a good color combination to have side-by-side due to color blindness.

Figure 4: You do not mention the abundance scaling here, since it appears from your gradient legend that it is a log10 scale (otherwise your numbers are meaningless). Why do you also use 8 colors/shades that are very close to each other (ie: many pairs of reused colors) instead of picking 8 very different colors for your phyla colors?

Tables S2/S3: You have the room in these tables’ headings to write out the full names of the taxonomic levels and not just the one-letter abbreviations.

Table S4/S5: The PD, ACE and Chao1 values can be rounded to the nearest whole numbers.

Figure S9: Are you sure each node represents a species, as you only label the genus and I don’t think I see duplicate genera names?

Figure S10: As this figure is huge, I would only display those functions with at least a certain threshold of mean. “Color of the circles, the relative higher values of the group” does not mean anything – what do you mean here?

---

## Round 0.2 · Minor Revisions

Please take into consideration the last comments raised by the reviewer

·

Basic reporting

No comment.

Experimental design

No comment.

Validity of the findings

No comment.

Additional comments

As the authors have addressed almost all of my previous comments, see below for the one exception (plus a new comment), I have no further revisions to add.

Specific comments:

Table 1: You still have the proportions listed as "rates", which is incorrect.

Figure S10: I forgot to mention it before, but as a minor comment (since this is for supplemental), I would switch your blue and brown colors since I automatically assumed that the brown was for the soil, but it was the inverse.

---

## Round 0.3 · accepted · Accept

Thanks for taking into consideration the very last changes.